# Identification of Ecological Sources Using Ecosystem Service Value and Vegetation Productivity Indicators: A Case Study of the Three-River Headwaters Region, Qinghai–Tibetan Plateau, China

**Xinyi Feng** [1,2], **Huiping Huang** [1,2,*], **Yingqi Wang** [1,2], **Yichen Tian** [1,2] **and Liping Li** [1]

[1] Aerospace Information Research Institute, Chinese Academy of Sciences, No. 9 Dengzhuang South Road, Haidian District, Beijing 100094, China; fengxinyi23@mails.ucas.ac.cn (X.F.); wangyingqi22@mails.ucas.ac.cn (Y.W.); liliping@aircas.ac.cn (L.L.)

[2] University of Chinese Academy of Sciences, No. 19(A) Yuquan Road, Shijingshan District, Beijing 100049, China

* Correspondence: huanghp@aircas.ac.cn; Tel.: +86-10-6488-9213

**Abstract:** As a crucial component of the ecological security pattern, ecological source (ES) plays a vital role in providing ecosystem service value (ESV) and conserving biodiversity. Previous studies have mostly considered ES only from either landscape change pattern or ecological function perspectives, and have ignored their integration and spatio-temporal evolutionary modeling. In this study, we proposed a multi-perspective framework for the spatio-temporal characteristics of ES by ESV incorporating landscape aesthetics, carbon sink characteristics, ecological quality, and kernel NDVI (kNDVI). By integrating the revised ESV and the kernel normalized difference vegetation index as a foundation, we employed the spatial priority model to identify ES. This improvement aims to yield a more practical and specific ESV result. Applying this framework to the Three-River Headwaters Region (TRHR), a significant spatio-temporal change in ecological sources has been observed from 2000 to 2020. This performance provided a reference for ecological conservation in the TRHR. The results indicate that this ecological source identification framework has reliable accuracy and efficiency compared with the existing NRs in the TRHR. This method could reveal more precise spatio-temporal distributions of ES, enhancing ecosystem integrity and providing technical modeling support for developing cross-scale spatial planning and management strategies for nature reserve boundaries. The framework proposed in our research could serve as a reference for building ecological networks in other ecologically fragile areas.

**Keywords:** ecological sources; ecosystem services value; vegetation productivity; Three-River Headwaters Region





## 1. Introduction

Constructed by ecological networks, ecological patterns play a crucial role in environmental restoration and regional sustainable development. Ecological source (ES) areas, defined as ecological patches in good ecological condition and providing high ecosystem service value (ESV) [1], are the first step and essential foundation in the construction of ecological networks [2,3]. Thanks to high-precision multi-source data products and well-developed analysis methods, "Source identification–Resistance surface construction–Ecological network generation" has become a globally classic paradigm for constructing ecological networks [4–6]. More researchers have focused on constructing resistance surfaces and exploring methods for ecological corridor identification [7–9]. The methods of ecological source identification have been limited to considering natural reserves, morphological spatial patterns [10–12], ecological importance, and sensitivity characteristics [13].

The most accessible and straightforward approach is the nature reserve method. Early studies have designated nature reserves as ecological sources [2,14,15]. However, constrained by their random distribution and subjective delineation, this method impedes objective differentiation based on diverse land use and landscape types. To compensate for the shortcomings of the nature reserves approach, some scholars have also employed the Remote Sensing Ecological Index (RSEI) to comprehensively identify ecological source areas [16]. However, it may not directly capture ecosystem functional patterns and characteristics of ecosystem services. Along with the developments of landscape morphology theory, Morphological Spatial Pattern Analysis (MSPA) is widely employed in ecological source extraction [10,11]. However, the theory concentrates on landscape connectivity while ignoring the community organization of the ecological space [17]. Under the background of ecological conservation [18], scholars are incorporating ecological importance and sensitivity features into the identification framework, considering ecological pressure and ecological quality [13], and using machine learning [19] to construct an ES identification framework. Although multiple processes in ecosystems are considered in this complex method, ESV, a crucial aspect of the ecosystem, has not received sufficient attention. Identifying ES based on ESV is another significant method. Diverging from the previously discussed methods, ESV offers a clear understanding of the coupling mechanism between humans and nature within an ecosystem, effectively illustrating the ecological function of the designated source areas [20,21]. Simultaneously, the incorporation of ESV enhances the visualization of the spatio-temporal evolution of ecosystems [22,23].

In summary, current ES extraction methods still face big challenges, and more in-depth and comprehensive identification methods are not yet available. First, most current research adopted a single perspective to identify ES. For instance, some studies focus on landscape change patterns or ecological functions in one aspect [20,24] but still need to fill the gap in different aspects. Second, connectivity and integrity are of great importance to the ecosystem [25], but it is challenging to consider the spatio-temporal continuous ES in recent identification methods. Hence, integrating various perspectives, including ecological functions and landscape patterns, and extending spatio-temporal evolutionary modeling approaches for ES identification present significant challenges and significance.

ESV assessments are a critical step in ES research that affect the accuracy of ES identification. The equivalent factor method has been frequently used in ESV assessments because it requires fewer elements and is applicable on a regional scale [26]. More improvements in the equivalent factor method are required for integrating perspectives of ecosystem service into the ES identification framework. Although most studies have used diverse components (e.g., NPP, precipitation) to correct the equivalent factor method [27,28], fewer studies have fully integrated ecological quality, carbon, and landscape aesthetic characteristics into the ESV revision. The inclusion of ecological quality elements into ESV can provide relevant insights for identifying ES spatio-temporal dynamic changes [29]. Carbon sequestration maintains healthy ecosystems and serves many benefits for human beings [30,31]. Meanwhile, ES also may provide nonmaterial contributions through recreation and aesthetic experiences but faces exceptional challenges to be integrated concretely and comprehensively into ES identification [32]. To thoroughly consider the multiple ecosystem characteristics, ecological quality, carbon sinks, and landscape aesthetics are urgently needed in an integrated adjustment of ESV to identify ES.

Vegetation productivity also plays an important role in ES identification [33]. As an essential ecosystem element, vegetation productivity stands out as highly responsive to climate change and human activities, offering a visible representation of shifts in ecological quality across a region [34,35]. Despite vegetation productivity being important and widely used in different ecosystem processes [36,37], integrating it into the ES extraction method has yet to be addressed. The normalized difference vegetation index (NDVI) offers extensive, multi-scale ecosystem monitoring and long-term vegetation trend information, being employed as a proxy for vegetation productivity [38]. However, it is also troubled by saturation effects for higher vegetation cover areas [39]. It is vital and urgent to consider

a more suitable vegetation productivity proxy factor comprehensively. In contrast to the seasonal oversaturation effect of NDVI, the kernel normalized difference vegetation index (kNDVI), proposed by Camps-Valls [40], is more suitable for modeling across multiple spatio-temporal scales. Moreover, it exhibits a certain level of correlation with the productivity parameters of plants [41,42]. Given that, it is worth considering incorporating kNDVI into the ecological source identification research framework.

Existing studies have discussed the importance of ESV and vegetation productivity in the ES identification framework, respectively, and the relationship between the vegetation condition and ecosystem service has been widely established [43]. However, the interaction and combination between ESV and vegetation productivity were often overlooked, which would be detrimental to understanding the overall ecological process interactions. Therefore, it is important to explore whether the combination of ESV and vegetation productivity can lead to more valuable outcomes of ES identification.

To solve the problems mentioned above, the primary objectives of this study were as follows: (1) Consider ESV as a pivotal factor in ES identification and revise it from multiple perspectives (landscape aesthetics, carbon sink characteristics, and ecological quality). (2) Integrate the multi spatio-temporal kNDVI into the process of identifying ecological sources. This inclusion allows for the quantitative identification of ES by considering spatio-temporal evolution, providing a comprehensive perspective for carbon sinks. (3) Identify ES using spatial priority models based on the normalized ESV and normalized kNDVI. (4) Demonstrate the spatio-temporal changes in ES by applying the established framework in the Three-River Headwaters Region (TRHR). The results could provide technical modeling support for developing cross-scale spatial planning and management strategies for nature reserve boundaries.

## 2. Study Area and Datasets

### 2.1. Study Area

Renowned as the birthplace of the Yellow River, Yangtze River, and Lancang River (Figure 1), the TRHR is dominated by grassland, forest, bare land, and water bodies. Meanwhile, the vegetation is mainly Alpine grasslands, which cover more than 60% of the region [44]. As the hinterland of the Qinghai–Tibetan Plateau, it boasts a diverse range of ecological species and extensive coverage of alpine meadows and alpine steppes [45]. The TRHR is unique and representative in the global natural ecosystem conservation and serves as a significant area to maintain global biodiversity at the levels of gene, species, and ecosystem [46]. It is also a crucial ecological barrier and Asia's biggest carbon sink zone [47].

The TRHR has low air temperatures and typical plateau climate characteristics [48]. The TRHR ecological environments are sensitive and fragmental, endangering its ecological security. Conducting an ES identification framework of the TRHR and delineating the ES pattern can address the existing research gaps and offer scientific insights for regional ecosystem conservation and sustainable development [49]. In turn, it promotes harmonious coexistence between human activities and the natural environment on the Qinghai–Tibet Plateau. So, we try to provide strategies for ecological restoration in this region.

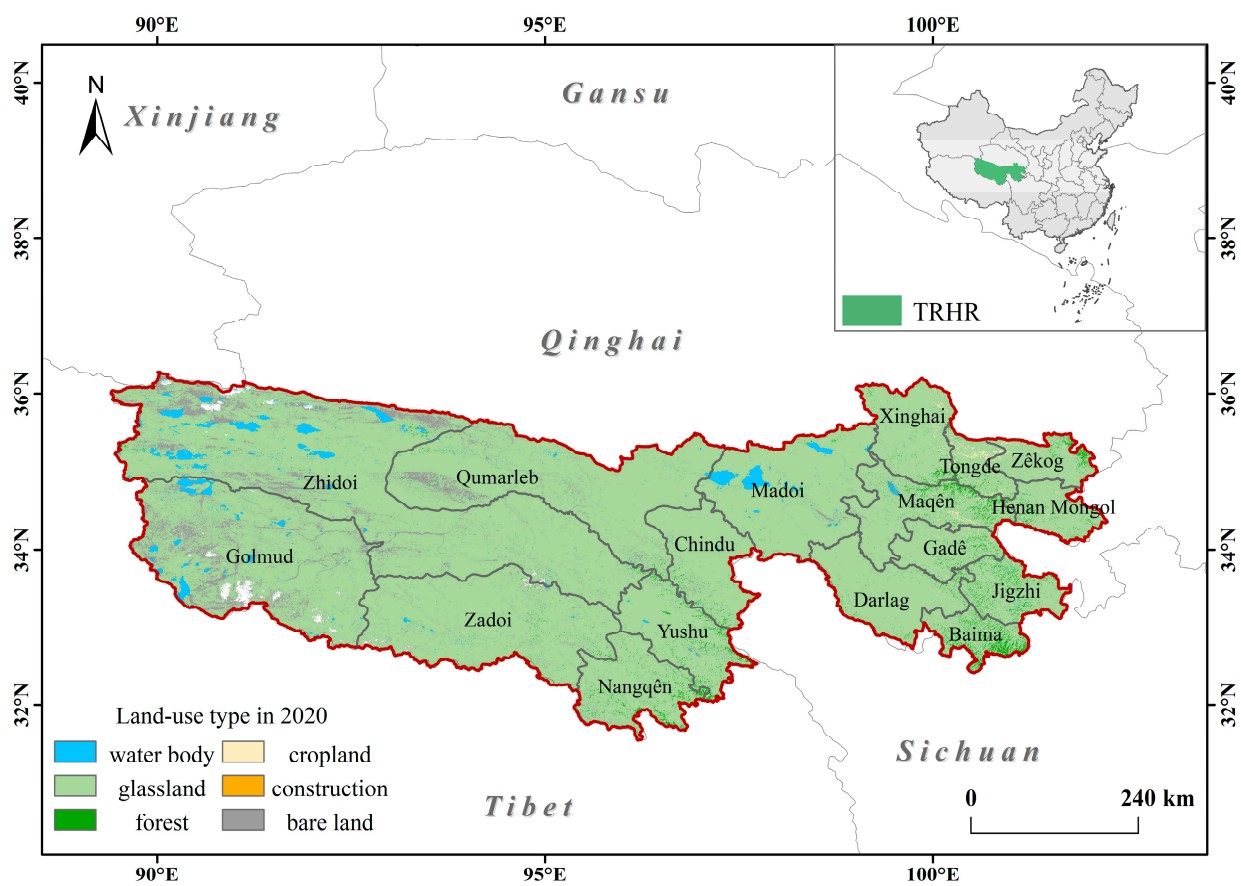

**Figure 1.** Land use types and the location of the study area.

### 2.2. Data Source and Processing

The ES identifications were based on quantitative remote sensing data, together with additional land-use data, productivity data, observations of the wildlife spatial distribution, and multiple socioeconomic components. To represent these factors and make sure that the analysis covers TRHR, we used different resolution data (Table 1). Considering data availability and the size of the study area, all spatial datasets are represented in a uniform coordinate system. To match all types of data, we resampled all raster data to a resolution of 1 km. The GlobeLand30 dataset [50] was recombined into six classes for subsequent ESV assessments. Based on the Google Earth Engine (GEE) platform, remote sensing data were processed annually for calculating the RSEI. Based on the Google Earth Engine (GEE) platform, remote sensing data were processed annually for calculating the RSEI. Specifically, low-clouded images from March to July were selected and retained all pixels that are categorized as "good" (QA = 0) or "marginal" (QA = 1) quality in the QA layer. Principal component analysis, water body masking, and declouding were among the other processes applied to the images [2]. The Yearly Net Primary Productivity data have been preprocessed with atmospheric correction, and the MODIS Reprojection Tool (MRT) software provided by NASA was used for data conversion, image stitching, and batch cropping using Python. The NPP data for the study area at five-year intervals from 2000 to 2020 were finally obtained. Precipitation data [51] and vegetation fraction coverage data [52] were processed on the timescale from monthly to annually and transferred from .NC format to .tiff format.

**Table 1.** Data used in this study and their sources.

| Dataset | Resolution | Dataset Name | Sources |
|---|---|---|---|
| Land Use [50] | 30 m | Global 30 m land-cover products with a fine classification system from 2000 to 2020 | https://data.casearth.cn/ (accessed on 24 March 2023) |
| Yearly Net Primary Productivity Data | 500 m | MOD17A3HGF | USGS EROS center https://www.usgs.gov/ (accessed on 15 January 2023) |
| Precipitation Data [51] | 1 km | 1 km monthly precipitation dataset for China (1901–2021) | https://poles.tpdc.ac.cn/ (accessed on 15 January 2023) |
| Fraction Vegetation Coverage Data [52] | 500 m | Aboveground biomass and vegetation cover data for the Qinghai–Tibet Plateau (1990–2020) | https://data.tpdc.ac.cn/ (accessed on 1 February 2023) |
| Remote Sensing Data | 30 m | MOD13A1V6, MOD11A2V6, MOD09A1 | Based on the GEE platform |
| Socioeconomic Data | - | «Qinghai Statistical Yearbook» «National Compilation of Agricultural Product Prices» «China Agricultural Yearbook» | http://tjj.qinghai.gov.cn/ http://www.stats.gov.cn/sj/ndsj/ (accessed on 15 January 2023) |
| Point of Interest Data | - | - | Gaode Open Platform |

## 3. Methodology

ESs were regarded as vital zones that provide multiple ecosystem services and vegetation productivity. An ES identification framework was conducted in this study and divided into three aspects: (1) assessing preliminary ESV by the equivalent factor method and calculating RSEI, Carbon Sink Index (CSI), and Landscape Aesthetics Index (LAI) to refine the ESV assessments; (2) estimating kNDVI (2000–2020) based on the GEE platform to describe spatio-temporal evolution; and (3) using different spatial priority rules, identify ES based on ESV and kNDVI. The concrete research framework is shown in Figure 2.

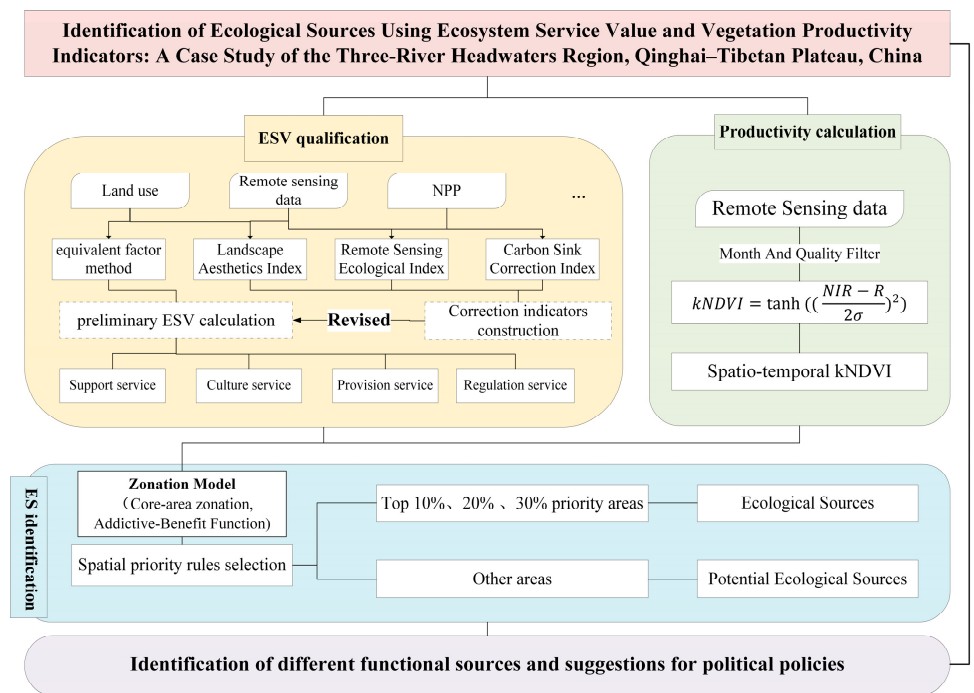

**Figure 2.** The workflow of ecological sources (ES) identification includes four phases: measuring ESV, calculating kNDVI, identifying different levels of ecological sources using the zonation model, and suggesting conservation policies.

### 3.1. Ecosystem Service Value Assessments

3.1.1. Quantification of the Ecosystem Services

ESV provides valuable insights into the mechanisms of ecosystem functioning and serves as one of the foundational datasets for identifying ES [53]. The equivalent factor method is one of the mainstream ESV assessment approaches and has been widely used [54]. ESV was categorized into supporting, regulating, provisioning, and cultural services based on the Millennium Ecosystem Assessment [55]. Combined with the equivalent factors table for terrestrial ecosystem services proposed by Xie [56], this study obtained the table of equivalent values for ecosystem services in the TRHR. Since the ESV provided by construction land is less challenging to measure accurately, this study does not assess construction land [57]. The total value of ESV was calculated according to the land use structure and different service types in the study area (Equation (1)):

$$\text{ESVs} = \sum_{i=1}^{n}\sum_{j=1}^{n} S_j E_{ij} \tag{1}$$

where ESV denotes the total value of ecosystem services in the study area (yuan), and $S_j$ means the land use area (hm$^2$) of the $j$ ecosystem service type. $E_{ij}$ represents the value of ecosystem services for the $j_{\text{th}}$ ecosystem service type of the $i_{\text{th}}$ land use type (Yuan/hm$^2$).

3.1.2. Revision of the ESV

Considering that an equivalent table is established for the whole China terrestrial ecosystem and only statically reflects the average ESV [27,58], targeted spatio-temporal revisions are needed when using it.

By synthesizing comparisons from earlier representative research and considering the TRHR's characteristics, we created the revised indicators of three dimensions—ecological quality, carbon emission and natural landscape aesthetics. Firstly, RSEI is used to reflect the spatial heterogeneity of ecosystems and characterize the influence of ecological quality in TRHR. It can capture the ecological quality rather than primal ESV. Secondly, the CSI is constructed to be relevant to ecosystems' capacities to cope with climate change and carry out carbon modification, which can complement the shortcomings of the primitive ESV. Lastly, most of the previous studies only considered the TRHR's ecological characteristics, ignoring its intangible cultural values. Thus, considering the scientific and cultural characteristics of the TRHR, the LAI is constructed to highlight the value of the TRHR's cultural services. In summary, these criteria represent the manifestation of three major components of the ESV (ecology, carbon emissions, and cultural characteristics).

(1) Remote Sensing Ecological Index.

Ecosystem quality is more accurately measured [2] based on the RSEI proposed by Xu [59]. Meanwhile, the relevant literature shows that RSEI demonstrates superior performance in the TRHR [60]. It is calculated as follows:

$$\text{RSEI} = f(\text{Greeness}, \text{Wetness}, \text{Heat}, \text{Dryness}) \tag{2}$$

where the Greeness, Wetness, Heat, Dryness are represented by different components assessed using remote sensing data. The RSEI was obtained by fusing the four major components and extracting the first principal component using Principal Component Analysis [61].

(2) Carbon Sink Revised Index.

As a crucial carbon-sensitive area of China's ecosystem, the TRHR has a vital carbon sink function. We qualified the CSI using Net Ecosystem Productivity (NEP) and Vegetation Fraction Coverage (VFC) indicators, which measure the level of carbon balance and can more reasonably highlight the value of carbon sinks provided by the TRHR.

$$CSI_i = \frac{1}{2} \times \left( \frac{NEP_i}{NEP_{mean}} + \frac{VFC_i}{VFC_{mean}} \right) \tag{3}$$

$CSI$ is the carbon sink revised index; $NEP_i$ is the NEP of study unit $i$ [44]; $NEP_{mean}$ is the average NEP of the study area; $VFC_i$ is the vegetation cover of study unit $i$; and $VFC_{mean}$ is the average vegetation cover of the study area. The explicit illustration of the expression of NEP can be found in [62], and specific descriptions of VFC are based on [52]. The calculation result for NEP is similar to the previous studies.

(3) Landscape Aesthetics Index.

A quantitative description of landscape patterns can effectively explore the internal mechanism of ecosystem action to reveal the ESV in the study area better. According to former research [63], Shannon's diversity index (SHDI), the Euclidean distance to places of interest, and naturalness were used to conduct the LAI indicator. The formula is presented below:

$$LAI_i = \frac{1}{3} \times (NT_i + NS_i + SHDI_i) \tag{4}$$

where $LAI_i$ is the LAI value of grid cell $i$; $NT_i$ is the naturalness of grid cell $i$ [64]; $NS_i$ is each grid's Euclidean distance to places of interest in TRHR; and $SHDI_i$ is the normalized SHDI of grid cell $i$. All three components were normalized to 1–100.

### 3.1.3. Revised ESV Calculation

After calculating the value of the preliminary ESV, the revised ESV in the study area can be calculated as follows:

$$ESV_{revise} = \sum_{i=1}^{6} \sum_{j=1}^{17} \sum_{t=1}^{n} R_{jt} \times E_{ij} \times S_{jt} \tag{5}$$

$$R_{jt} = \text{Normalize}(LAI_{it} \times CSI_{it} \times RSEI_{it}) \tag{6}$$

where $i$, $j$, and $t$ represent the ecosystem category, service type category, and the number of image elements, respectively. $ESV_{revise}$ is the revised ecosystem service value; $R_{it}$ is the revision index after normalization; $E_{ij}$ is the grid's unrevised ecosystem service value; and $S_{jt}$ is each grid's area. $LAI_{it}$ is the Landscape Aesthetic Index, $CSI_{it}$ is the Carbon Sink revised Index, and $RSEI_{it}$ is the Remote Sensing Ecological Index.

After the above modifications, the dynamic results of the ESV assessment in the TRHR were obtained, which considered more comprehensively the ecological characteristics, carbon sinks, and landscape aesthetics at the grid scale of the TRHR and could further improve the accuracy and comparability of ES identification.

### 3.2. Kernel Normalized Difference Vegetation Index Calculation

kNDVI can offer substantial support in investigating regional carbon sinks. To incorporate the impacts of carbon sequestration in our ES identification methods, we take kNDVI into our framework. In previous studies [65], it has been demonstrated that kNDVI can effectively alleviate the saturation problem and showcase high robustness in performance compared to NDVI. It is defined by adopting the RBF kernel function, recovers all higher order differences between the NIR and red reflectance bands, and when an appropriate kernel function is used, the index comes in a relatively simple and practical expression [40,66]. The simplified equation is as follows:

$$kNDVI = \tanh\left(\left(\frac{NIR - R}{2\sigma}\right)^2\right) \tag{7}$$

where NIR and $R$ are the reflectance in the NIR and the red bands, respectively, and $\sigma$ is a kernel length scale parameter that must be specified in each application. The application of the kernel function and parameter allows kNDVI to deal with the seasonal saturation effects and reflect complex vegetation phenology [40].

In this study, kNDVI was calculated on the GEE platform using the MODIS dataset. Year-by-year kNDVI values from 2000 to 2020 at a location (x, y) were used to fit this series

with the linear regression model, and the slope was used to reflect the trend of kNDVI at that location [67].

### 3.3. Ecological Source Identification

Balancing numerous considerations and ensuring the continuity of ESs is vital for ES identification, and previous studies have achieved this goal using ZONATION models in different aspects [5,53,68]. Thus, accounting for the interaction of ecosystem service and vegetation productivity, we integrated normalized ESV and normalized kNDVI as input data for spatial conservation prioritization in the decision support software ZONATION (v.4.0). Zonation software (v.4.0) assumes that ecological priority values of all areas need to be protected and range from 0 (cannot provide ecological value, e.g., urban areas) to 1 (provide multiple ecological value areas) [69]. The order of deletion reflects the relevance of the units to conservation: regions deleted first have a lower priority, while those retained longer have a greater priority [70]. Models iteratively eliminate the pixel with the most minor contribution to the conservation objective [6] and generate priority rank maps as its primary output.

Calculating the marginal removal loss, also known as the cell removal rule, is directly related to the order of raster removal during the model run [71]. The Core-area zonation (CAZ) and Addictive–Benefit Function (ABF) rules are mainly used in previous studies [72,73]. In the CAZ method, the removal of cells is based on a value called the removal index $\delta_i$ [74]. The formula is as follows:

$$\delta_i = max_j \frac{q_{ij} w_j}{c_i} \tag{8}$$

where $\delta_i$ represents the minimum marginal loss of the biological value of the cells; $q_{ij}$ is the proportion of the remaining distribution of feature $j$ located in cell $i$ for a given set of sites (the set of cells remaining); and $w_j$ represents the weight or priority assigned to species $j$, while $c_i$ stands for the cost of incorporating cell $i$ into the reserve network. The cell with the lowest $\delta_i$ value is selected for removal.

In contrast to the CAZ rule, the ABF rule considers the proportions of all features in a given cell, rather than only the greatest value. All species in this rule can compensate for one another [74]: losing some representation for one species results in a loss of value for that species, but the loss can be at least partially offset by higher representation for other species elsewhere. The formula is as follows:

$$\delta_i = \frac{1}{c_i} w_j \sum_j \Delta V_j = \frac{1}{c_i} w_j \sum_j \left[ V_j(q_j) - V_j(q_j - i) \right] \tag{9}$$

where $q_j$ is the representation of feature$_j$ in the remaining set of sites, and $(q_{j-i})$ indicates the set of remaining cells minus cell $i$. Here, $w_j$ is the weight of the feature $j$ and $c_i$ is the cost (or area) of planning unit $i$. As in CAZ, the cell with the smallest $\delta_i$ value will be removed.

To better protect integrated and single ecological elements, we integrated the conservation priority areas under CAZ and ABF rules to identify ES. With the CAZ and ABF methods, areas recognized as not in the top 30% are assigned as potential areas. Additionally, the lower value will be selected with multiple superimposed spatial priority rules.

Considering that the zonation algorithm produces hierarchical priority maps and considering previous studies [75], we extract the top 10%, 20%, and 30% of conservation priority areas as ESs and other areas as potential ESs, respectively. The logic for the classification is as follows: (1) the top 10% (the raster value of 0.9–1) priority area, distinguished by its capacity to provide the highest ecological function value; (2) the top 20% area (the raster value of 0.8–0.9), acting as a buffer for the top 10% priority zone; (3) the top 30% (the raster value of 0.7–0.8) priority area, serving as a linkage between ecological sources and the potential area; and (4) the remaining potential areas, capable of transforming into ecosystem sources through environmental improvements.

## 4. Results

### 4.1. ESV and KNDVI of the TRHR Present Significant Spatial Heterogeneity

The total ESV from 2000 to 2020 tends to increase and then decrease (Figure 3). Among the four major types of service types, regulating value occupies a dominant position and has the most apparent fluctuation; support services are the second most important, with a slower degree of change; and cultural services and supply services have provided a more stable ESV over the past two decades and account for a smaller proportion of the total.

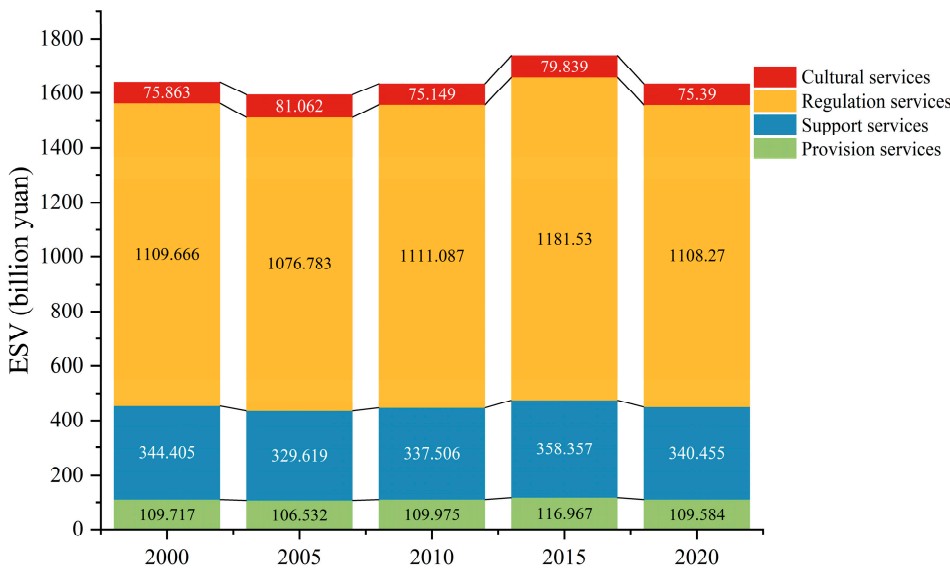

**Figure 3.** Temporal changes in the four ecosystem services in the TRHR.

In general, the ESV of the TRHR demonstrates spatial heterogeneity, with a stable spatial layout of low-value clustering and high-value dispersion (Figure 4). An overall trend of decreasing from southeast to northwest was observed, which is closely related to the characteristic of increasing temperature and humidity from northwest to southeast in the TRHR [76]. As shown in Figure 4f, the ESV in the southeastern TRHR declined as a result of human-caused ecological land degradation. An increasing trend is mainly shown in the middle of the TRHR.

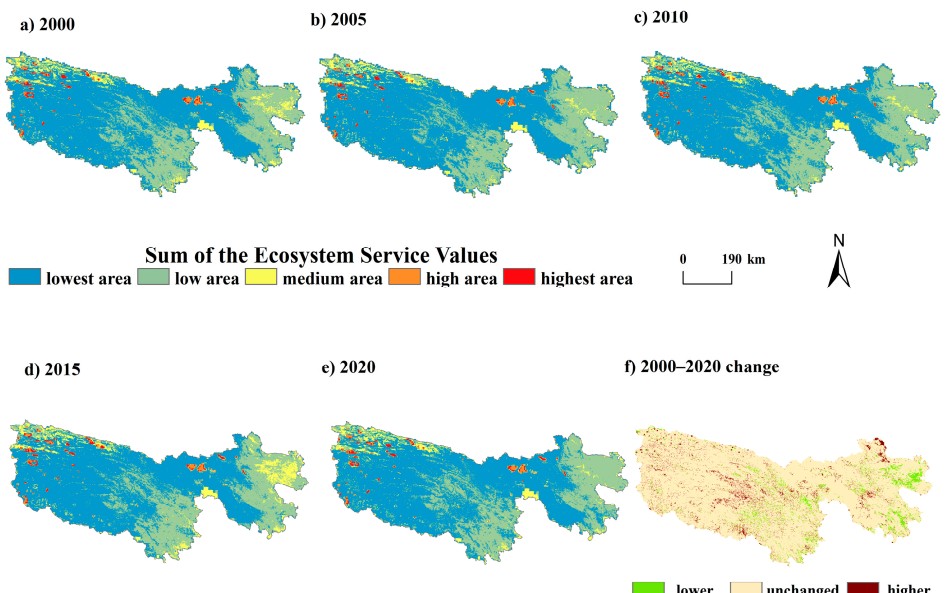

**Figure 4.** Spatial distribution (**a**–**e**) and spatio-temporal changes (**f**) in ESV from 2000 to 2020.

Specifically, the highest-value areas are reflecting the continued clarification of ecosystem boundaries. They are mainly distributed in the central part of the region, Zaling Lake, Eling Lake, and the northwestern part of the Hoh Xili Lake, Xijinwulan Lake, Sun Lake, and other alpine river and lake areas. As multifunctional ecosystems, these rivers and lakes play essential roles in water conservation, soil conservation, carbon sequestration, and oxygen release; so, the watershed areas dominate the ESV in different periods. The areas with high and medium values are scattered in the eastern and southeastern parts of the TRHR, and the trend of low values evolving to medium values in the eastern part gradually increases as time passes. The eastern part is the Yellow River Basin, and the southern part has a more favorable climate, where ecosystems such as alpine meadows and alpine grasslands are widely distributed and can provide stable high-value functions such as carbon sequestration and oxygen release, climate regulation, and environmental purification. Therefore, these zones have higher ESVs relative to the northwest.

The low-value areas are mainly concentrated in the northwest and part of the central areas of the region, of which the Yangtze River source area occupies a higher proportion, including the core conservation area and the Qinghai Hoh Xil.

The kNDVI from 2000 to 2020 is shown in Figure 5. There was significant spatio-temporal heterogeneity of the kNDVI in the TRHR. From 2000 to 2020, the kNDVI increased gradually from northwest to southeast (Figure 5a–e) with strong longitudinal zonality. Regarding the spatial distribution features, the high values are primarily concentrated in the southeastern region, where mountain forests and alpine meadows are widespread. In the northwestern part, characterized by glaciers and desert bare land, vegetation cover is low and discontinuous. The linear regression model was used to detect trends of changes in the kNDVI. The pixels with an increasing trend comprised 34.5% of the entire research area, which was much larger than those with a decreasing trend (3%) because of vegetation restoration from barren or sparsely vegetated areas to grassland.

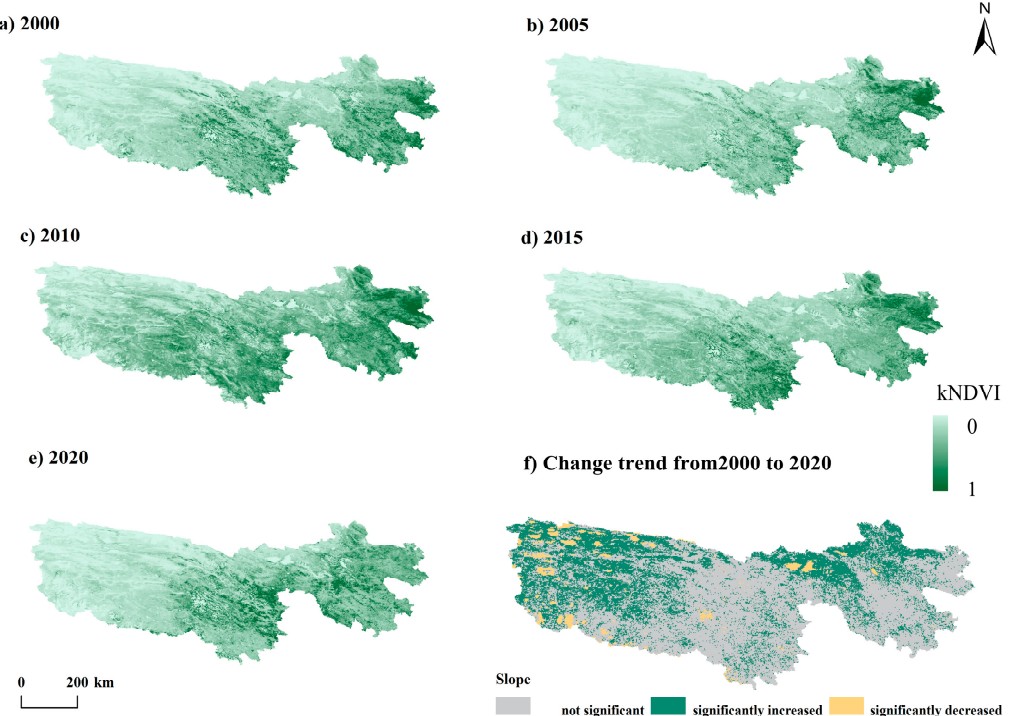

**Figure 5.** Spatio-temporal dynamic changes in the kNDVI from 2000 to 2020 in the TRHR.

### 4.2. Ecological Source Identification Model Performance Analysis

As mentioned in the methodology, ESs within the TRHR from 2000 to 2020 were determined by applying a zoning model built upon the CAZ and ABF selection criteria, considering the overlapping results of both methodologies. As shown in Figure 6, ES identification results are diverse between different ecological priority rules. The CAZ method extracted source sites that are ecologically preferred. However, the resulting patches were smaller and exhibited a more random distribution. In contrast, the ABF method produced more continuous results but overlooked the significance of water bodies as crucial ecological source providers. So, it is necessary to combine the two extraction methods.

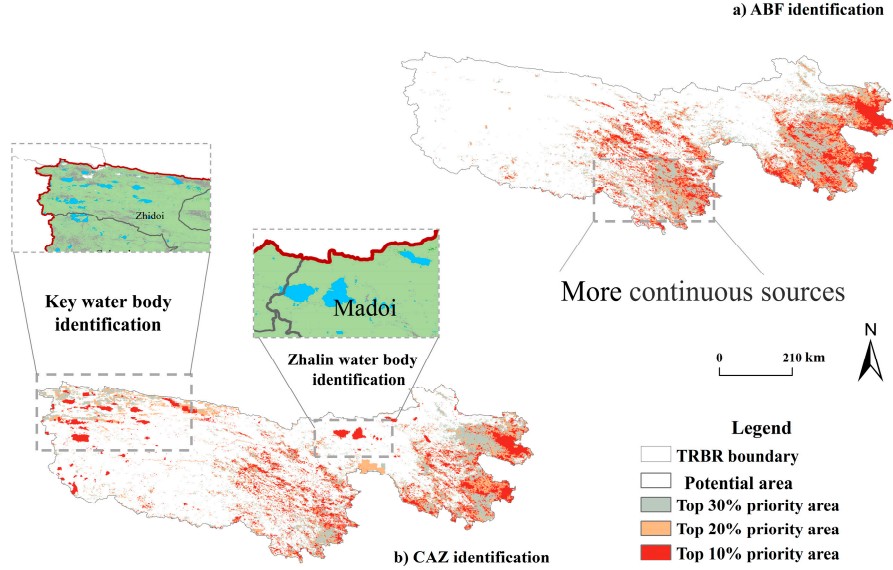

**Figure 6.** Ecological sources identified from the (**a**) ABF and (**b**) CAZ methods in the TRHR.

Based on Table 2, the temporal changes reveal a sustained presence of potential areas for 20 years. The conversion of areas into ES sites shows an increasing trend over time, with 20% of the total TRHR being recognized as ecological source areas. These identified ESs provide a high ESV and carbon sinks. From 2000 to 2015, the ecological patches fluctuated insignificantly and primarily had an interconversion of the top 20% and 30% of areas. Notably, the top 30% of priority areas exhibit a gradual decrease, with about 70% of the areas transforming into the top 20% of ecological source sites from 2015 to 2020. Meanwhile, the top 10% of total source areas remain relatively stable, showing minimal and non-significant changes.

**Table 2.** Ecological sources in the TRHR from 2000 to 2020.

| Year | Potential Area (km²) | Top 30% Area (km²) | Top 20% Area (km²) | Top 10% Area (km²) |
|---|---|---|---|---|
| 2000 | 319,917 | 32,061 | 29,836 | 29,775 |
| 2005 | 318,850 | 33,349 | 30,081 | 29,288 |
| 2010 | 312,863 | 35,624 | 32,622 | 30,480 |
| 2015 | 315,171 | 35,437 | 30,018 | 30,963 |
| 2020 | 309,338 | 6627 | 65,781 | 29,749 |

Regarding spatial distribution depicted in Figures 7 and 8, ES exhibits distinct characteristics. In the northwest, there is a discrete and sparse arrangement, while a more consolidated, block- and surface-like distribution is evident in the southeast. From 2000 to 2010, the spatial distribution of ES was stable. The top 10% of areas are mainly distributed in Henan Mongol Autonomous Prefecture and Jigzhi County, which are sporadically dis-

tributed in the central regions. The top 20% of ESs are spread in a ring around the top 10% of ES areas.

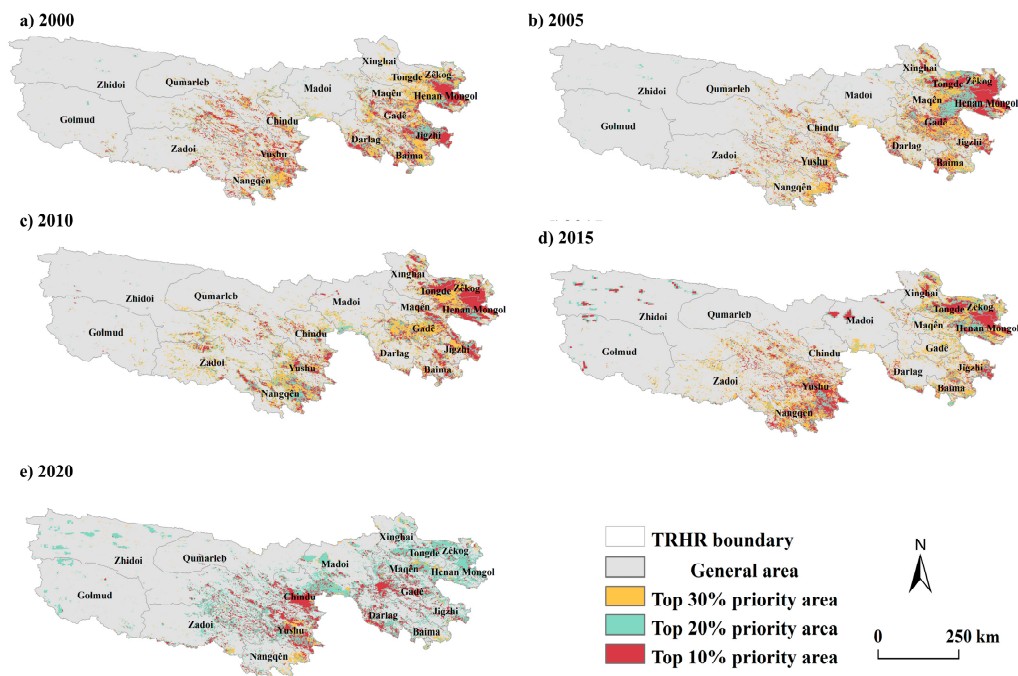

**Figure 7.** Spatio-temporal patterns of priority areas of the TRHR from 2000 to 2020.

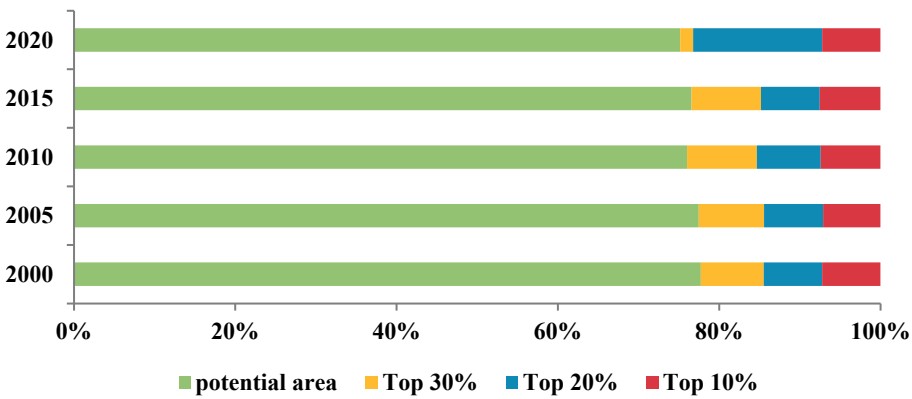

**Figure 8.** ES composition and the temporal changes in the TRHR.

Since 2010, Zhalin lakes and others in the northwest have transitioned gradually from potential areas to the top 30% of areas, while the top 10% of ES patches have decreased in the central region. The establishment of the Sanjiangyuan National Park in 2012 has played a positive role in maintaining and cultivating ecological sources. A discernible trend in some southeastern areas indicates an initial increase followed by a subsequent decrease from 2000 to 2020. Notably, in 2020, certain areas classified initially as the top 10% have been downgraded to the top 20%, but other areas are still showing an improving trend.

As depicted by Xu et al. [77], the grassland coverage and productivity increased while grassland degradation was controlled and desertification was prevented after the first phase of the Ecological Project (2005–2012) in the TRHR. It is also reflected in our results. Although the potential area still cover huge areas in the TRHR, the top 30% of areas, which mainly consist of grassland, were transferred into the top 20% of areas gradually. According to Ning et al. [78], the restoration area far increased during the implementation of the second phase of the Ecological Project. Thus, it is not hard to understand that the top 30% of ESs

will change into the top 20% of ESs. However, the degraded top 10% of Ess are distributed in eastern part of the TRHR. These degraded areas are mainly dominated by human activities, which will offset the temporary ecological restoration, causing ES degradation.

## 5. Discussion

### 5.1. Implications of the Ecological Source Identification Framework

Selecting critical patches with ecological advantages is the basis for enhancing the connectivity and integrity of ecosystems [53]. However, the current identification mechanism fails to model and quantify the continuity of ecosystems. Moreover, inadequate recognition of complex ecosystem spatio-temporal evolution and large-scale zoning management limits the effect of planning implementation [79]. Specifically, previous studies have only considered it from a single spatial or temporal perspective and ignored the carbon sinks and ecological service characteristics. Compared with the related ES identification work, our study combines multiple ecosystem characteristics and carbon sinks into ES identification workflows, which effectively complements current ES research. To enhance the original ESV assessment method for the TRHR, we incorporate landscape aesthetics, carbon sink characteristics, and ecological quality considerations. This refinement aims to yield a more practical and site-specific ESV result. Additionally, we calculate the unified vegetation index, kNDVI, to enhance the representation of spatio-temporal scale carbon sink characteristics within the research zones. Moreover, we employ spatial priority configuration tools and integrate different cell removal rules to extract ecological sources, categorizing them into three distinct classes. Therefore, the multiple ecosystem service values and ecological integrity have been fully considered when assessing ecological sources.

### 5.2. Comparison of the Proposed Method and Previous Methods

Compared with the ESV assessments in previous studies [44,80], our revised results were consistent with the results: "*ESV decreasing from southeast to northwest in the TRHR*". To validate the simulated ES results, we compared the ES with Li et al. [75]. Our results are similar to the previous results in the southeast and middle zones, indicating that the identification results are reliable. However, it is noteworthy that some differences were observed in parts of the northwest. We believe these differences may be attributed to variations in ESV assessment methods and data resolution, and our identification results reveal more intricate patterns.

Specifically, to understand the difference across conservation objectives, we identified and mapped ESs that would be missed if conservation efforts were to focus solely on the ESV or kNDVI. We have considered three scenarios: ① only considered the kNDVI, ② only considered the ESV, and ③ combined the ESV and kNDVI into identification. We used the same spatial priority methods and analyzed the spatial distribution and temporal differences in three scenarios. From a spatial perspective, there are also some benefits and drawbacks to ES identification using the kNDVI or ESV, respectively (Figure 9). Identifying ESs through the kNDVI can reveal the ecological contribution of the Yangtze River Headwaters Basin in the southeast of the TRHR. However, the kNDVI is unable to identify the lakes in ESs, due to the theory of vegetation index (the red solid box in Figure 9). Meanwhile, ES identification emphasized the northwest ecological function but neglected the middle areas (the black dotted box in Figure 9). To detect ES, it is crucial to integrate the kNDVI and ESV. Thus, we decided to combine the ES identification results from both the kNDVI and ESV to exact more comprehensive ESs.

Given the difficulty in obtaining identification results from other authors, we quantitatively present a comparison of source site results using only the ESV (the traditional method employed by previous studies) and incorporating the kNDVI (the method used in our study). According to previous studies [53], we used landscape metrics calculated by Fragstats4.0 to compare the ESs identified using different approaches.

The Split Index represents the total degree of landscape dispersion, whereas the Division Index represents the degree of landscape separation. The high Split and Division

Indices indicate terrain fragmentation in research areas, which does not facilitate energy flow between ESs. As depicted in Figure 10, the Split Index and Division Index of ESs identified by the proposed method were significantly lower than those of the traditional method.

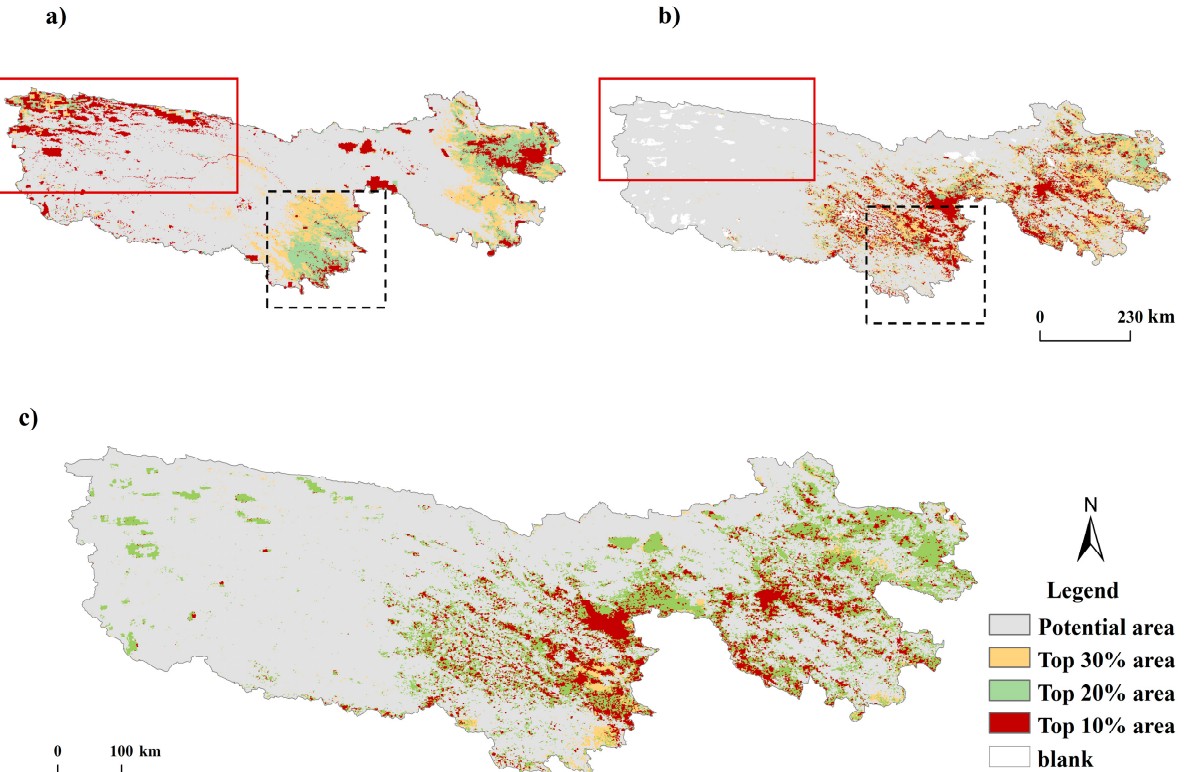

**Figure 9.** Ecological source identification differences between (**a**) ESV and kNDVI indicators, (**b**) only the kNDVI indicator, and (**c**) only the ESV indicator.

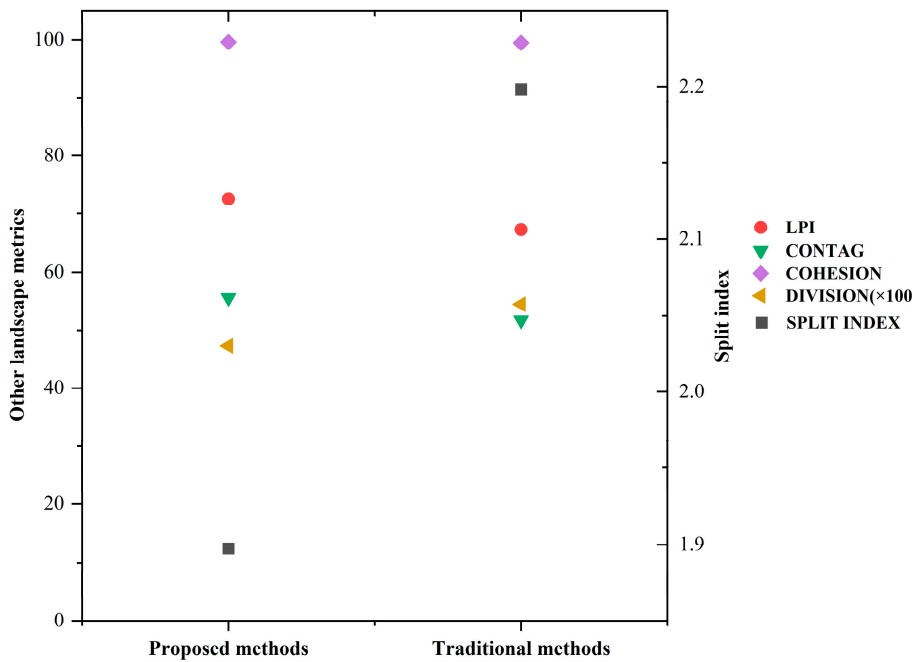

**Figure 10.** Landscape metrics of ecological sources identified by different approaches.

The Largest Patch Index (LPI) characterizes the strength of landscape fragmentation, heterogeneity, and human disturbance [81]. A high LPI can reflect the stability of the total region. The Patch Cohesion Index (COHESION) and the Contagion index (CONTAG) further indicate natural connectivity and a concentrated distribution of ESs. Stronger ES identification efficiency and accuracy were demonstrated by the high LPI, COHENSION, and CONTAG. ESs identified by our method were slightly higher than the traditional way, which indicates that the degree of ES aggregation improved following identification by the proposed method, as well as the existence of dominant patch types with high connectivity in the landscape [82].

### 5.3. Management Implications Based on the Identified Ecological Sources

We can easily find that most areas in the northwest have a stable low ESV. Learning from land use in the TRHR (Figure 11b,c), the region has complex land use types with typical alpine plateau characteristics ranging from alpine desert steppe to cold desert steppe and alpine meadows with many high lakes, marshes, and snow-capped glaciers. The world heritage site Qinghai Hoh Xil Natural Protection area is located here, which is called the 'core center Conservation Area' in the Sanjiangyuan National Park planning policy. So, it is our goal to improve the environment and preserve the authenticity of ecosystems in the future.

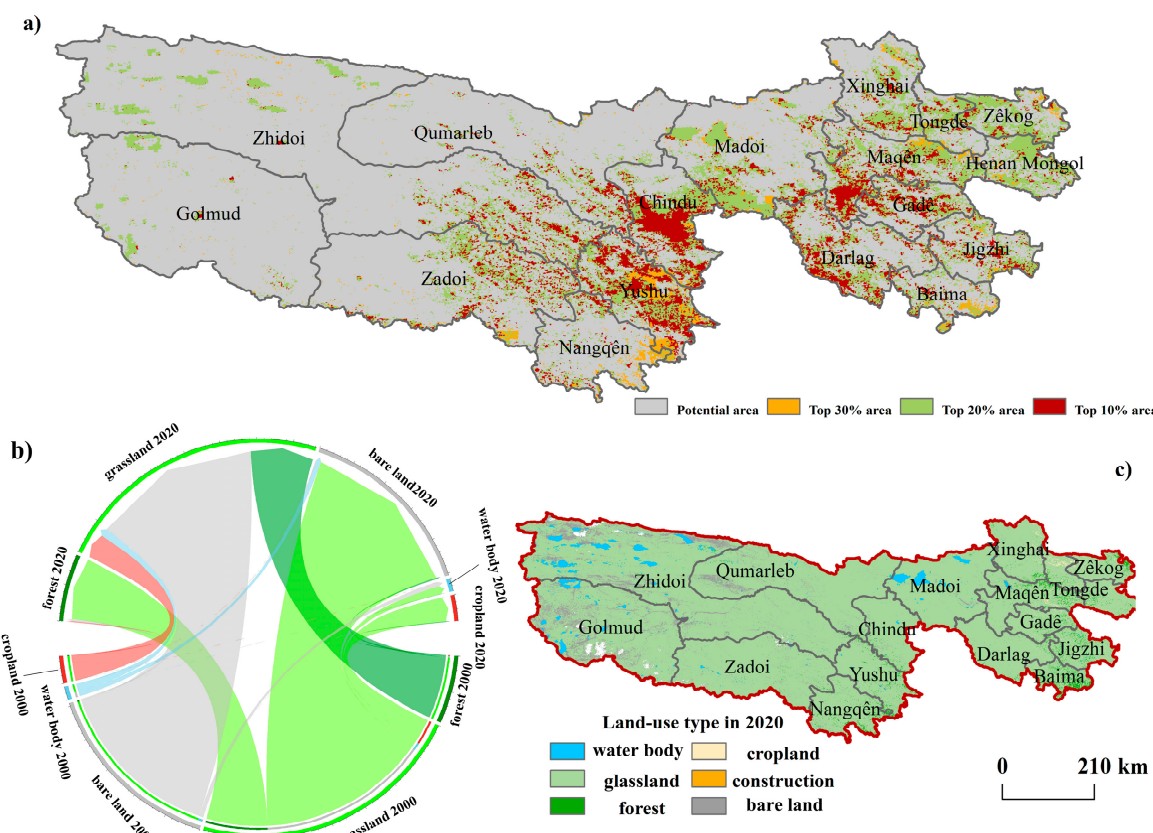

**Figure 11.** Identified ESs (**a**), the land-use transition from 2000 to 2020 (**b**), and the land-use spatial distribution in 2020 (**c**) in the TRHR. (Notably, construction was not considered in Figure 11b due to its small proportion.)

Nature reserves (NRs) are regarded as pillars of biodiversity conservation [34], contributing to the enhancement of suitable wildlife habitat quality. Considering the relevant policies and conservation programs, we compare ESs extracted in our study with the NRs proposed by the policies and the main wildlife spatial distribution points of the TRHR [83].

As illustrated in Figure 12, it is evident that specific ESs align with the designated protected areas, with over 50% of wildlife distributed within these areas. Furthermore, some wild animals exhibit activity near these ecological sources. The findings closely reflect the reliability of the ES results, underscoring their suitability for delineating priority conservation areas in the TRHR.

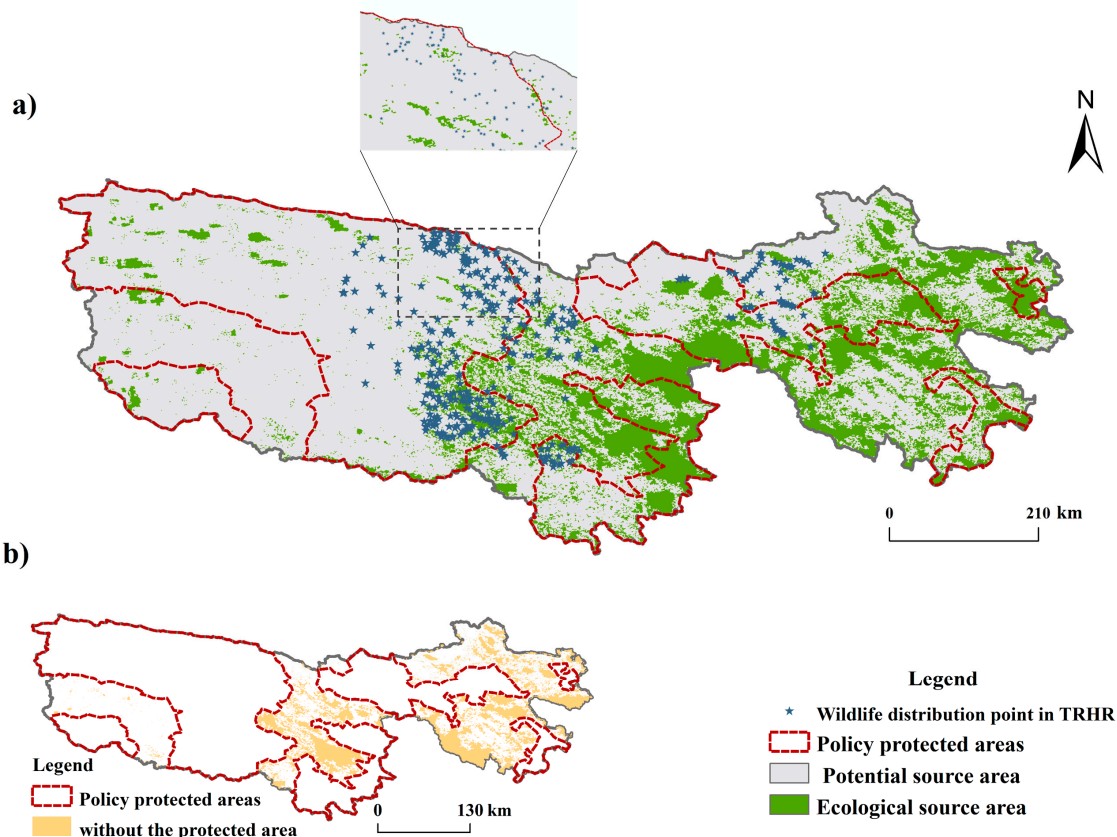

**Figure 12.** Comparison between extracted ecological source areas and policy-designated protected areas, with the wildlife spatial distribution in 2017 (**a**) and different areas between nature reserves and ecological source sites extracted in this study (**b**).

We found that 48,102.145 km² of ecological sources are located out of the NRs. This is mainly because of the difference in the definition approach between NRs and ESs. The setup of NRs considers mainly the distribution of species diversity [46], which may neglect the ESV to a certain extent. The existing NRs are not entirely sufficient to support species activities in the TRHR. Due to the long migration paths of wildlife and the wide home range, local ungulate species, i.e., *Equus kiang* and *Canis lupus*, are distributed in the edge or even out of NRs, and so the existing NRs are not entirely sufficient to support species activities [84,85]. Hence, the results of our study can provide different insights for refining the delineation of NRs.

Identification of ESs in the TRHR can help the government formulate policies (e.g., ecological restoration projects) and utilize the multiple benefits delivered by ESs. The governments should realign the NRs to meet the distribution of ESs in addition to biodiversity. For example, eastern ES patches could be included in the NRs in the future. Given the peculiar ES spatial geographic position of the TRHR, policymakers must establish hierarchical and zonal management for achieving multiple international agreements and targets [86].

*5.4. Limitations and Future Work*

Several limitations were noted in our ES identification framework. Due to the availability of datasets, we calculated the ESV using the equivalent factor method. However, results could vary across different methods and cause differences in ES extraction results, as described in the Discussion section. A more meaningful approach involves contrasting the disparities in the ESV derived from diverse calculation methods that could be developed for ES extraction in the future.

Additionally, although previous studies have achieved the ideal result using spatial prioritization tools [87] on a different scale, ES extraction must be combined with the trade-offs and synergies between ecosystem services and other aspects of in-depth research. We hope that subsequent research can improve this research with trade-off models and other methods in the future.

Furthermore, although the kNDVI may represent primary productivity and further help us understand carbon sinks to some extent [8], there are some discrepancies between it and the other parameters of terrestrial ecosystems (e.g., GPP, NPP) due to the potential spectral saturation effect. The kNDVI alone may not be sufficient to detect ecological source changes driven by carbon conditions. Research to enrich the identification framework with critical parameters for characterizing terrestrial ecological processes could be conducted.

Despite these limitations, our findings provide valuable insights into applying a multi-perspective approach to ecological source identification.

## 6. Conclusions

In this study, we quantified the ESV and kNDVI and integrated them into the ES identification framework. Modeling was completed using a Zonation model combined with two spatial priority rules, extracting ESs at different levels based on various thresholds. This framework was applied to the TRHR from 2000 to 2020. The results indicate that this method could reveal more precise spatio-temporal distributions of ESs, enhancing ecosystem integrity and providing technical modeling support for developing cross-scale spatial planning and management strategies for nature reserve boundaries.

Firstly, the ESV was taken as an essential factor in ES identification, and multiple perspectives were considered in the ESV calculation, which solved the problem of focusing on a single perspective in the previous ES identification method. It could be concluded that most of the ESV showed a spatio-temporal increasing trend, with the southeast part gently decreasing.

Secondly, representing vegetation productivity, the kNDVI was incorporated into the ES identification process, ensuring that the identification outcomes accurately capture the spatio-temporal dynamics of vegetation and carbon sinks. We found that the kNDVI of the TRHR showed longitudinal zonality and increased largely from 2000 to 2020.

Thirdly, the ES identification framework is applied to the TRHR, and the ES has a certain degree of reliability. According to the proposed identification framework, different threshold ecological sources were detected, and most areas were distributed in the southeast. A significant spatio-temporal change in ecological sources has been observed since 2000 in the TRHR. Compared with the existing NRs in the TRHR, the ecological source identification framework had reliable accuracy and efficiency. The spatio-temporal variations of ecological sources provide a reference for ecological conservation in the TRHR. The framework proposed in our research could serve as a reference for building ecological networks in other ecologically fragile areas, as well as a step towards ecological security pattern construction. More thorough research should be concentrated on spatio-temporal distribution drivers of ESs and mechanisms of transition in the future. Meanwhile, the distinction between ESs and NRs should be examined to generate the highest benefits for different protection purposes. Additionally, a global perspective of ES identification needs to be expanded.

All acronyms in this paper are listed in Table 3.

**Table 3.** The acronyms comparison table used in this paper.

| Acronym | Full Name | Explanation | References |
|---|---|---|---|
| ES | ecological source | Ecological source areas represent continuum patches that are important for biodiversity, ecosystem services, and regional ecological security, or they have important radiative functions. | [1] |
| ESV | ecosystem service value | The direct and indirect benefits to human welfare offered by ecosystems. | [28] |
| RSEI | Remote Sensing Ecological Index | A composite ecological indicator that incorporates four key parameters—greenness, humidity, dryness, and heat—to evaluate the ecosystem quality. | [88] |
| MSPA | Morphological Spatial Pattern Analysis | An imaging method based on grid pixels of land use in the study area for calculation, identification, and segmentation | [10,89] |
| kNDVI | kernel Normalized Difference Vegetation Index | A vegetation index based on kernel methods expressed in terms of the spectral channels. | [40] |
| TRHR | Three-River Headwaters Region | The source of the Yangtze River, the Yellow River, and the Lantsang River. | [90] |
| GEE | Google Earth Engine | An image dataset processing cloud platform | [91] |
| CSI | Carbon Sink Index | A carbon sink revision index | - |
| LAI | Landscape Aesthetics Index | A landscape aesthetics revision index that consists of naturalness, the Shannon–Wiener diversity index, and the Euclidean distance to places of interest. | [63] |
| NEP | Net Ecosystem Productivity | The difference between net primary productivity (NPP) and soil heterotrophic respiration ($R_h$). | [92] |
| VFC | Vegetation Fraction Coverage | One of the most important indicators for measuring surface vegetation cover. | [93] |
| CAZ | Core-area zonation | A spatial priority rule that tries to retain core areas of all species. | [69] |
| ABF | Addictive–Benefit Function | A spatial priority rule that tries to retain core areas of all species. | [69] |
| NRs | Nature reserves | Pillars of biodiversity conservation. | [34] |

**Author Contributions:** Conceptualization, X.F. and H.H.; methodology, X.F., H.H., Y.W. and Y.T.; validation, X.F. and Y.W.; formal analysis, X.F. and H.H.; investigation, X.F. and Y.W.; writing—original draft preparation, X.F.; writing—review and editing, H.H., Y.W. and L.L.; visualization, X.F. and L.L.; supervision, Y.T.; funding acquisition, H.H. and Y.T. All authors have read and agreed to the published version of the manuscript.

**Funding:** This research was funded by the MAJOR PROJECT OF CHINESE HIGH-RESOLUTION EARTH OBSERVATION SYSTEM from National Geospatial Information Center, grant number: 00-Y30B01-9001-22/23.

**Data Availability Statement:** All the data products in this study can be downloaded for free from the website links attached in the main text.

**Acknowledgments:** The authors gratefully thank the editors and anonymous reviewers for their valuable advice in improving this manuscript.

**Conflicts of Interest:** The authors declare no conflicts of interest.

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
