# Peer review of "Identification of Ecological Sources Using Ecosystem Service Value and Vegetation Productivity Indicators: A Case Study of the Three-River Headwaters Region, Qinghai–Tibetan Plateau, China"

_remotesensing, doi:10.3390/rs16071258_

Round 1
Reviewer 1 Report
Comments and Suggestions for Authors
Dear authors,
I think your study is interesting.
I have marked the small word mistakes I saw in the PDF and the points that are not in accordance with the template. You will see my notes.
The first PDF I downloaded did not have the "figures". Since I made my corrections in that PDF, I continued with it.
Your figures are generally nice and clear, except Figure 9b :)
As I mentioned in the PDF, if you use spatio-temporal instead of spatial-temporal, it will make more sense and will be used in this way more often.

Minor corrections to English are needed.
Author Response
We truly appreciate the reviewer's specific and constructive suggestions on our
manuscript. According to your helpful ideas, we have made considerable corrections in our draft. Please see the attachment.

Reviewer 2 Report
Comments and Suggestions for Authors Even though it seems to carry out a new application of ecological source identification using ecosystem services and vegetation productivity indicators, the study is not innovative enough, and the difference from previous studies seems to be adding only a new indicator for calculation. And the combination of the two indicator dimensions (i.e., ecosystem services and vegetation productivity) does not seem to make sense. The study is likely to become a repetitive experiment if the authors fail to explain the advantages and significance of combining the two metrics as well as the innovation of the study. At the same time, the study was analyzed with a single case only. It is worth pondering whether the analysis with a single case is sufficiently representative and under international influence. Comments on the details of the article are as follows: 1.Methodological Clarity: The criteria for the selection of indicators and the steps of pre-processing need to be explained in more detail. 2.Comparative Analysis: A more thorough comparison with existing methodologies could highlight the unique contributions more effectively of the study. This includes discussing the advantages and potential drawbacks ofthe proposed approach in relation to traditional methods. 3.Figure 2:In the technical route, the juxtaposition of "equivalent factor method, Landscape Aesthetics Index, Remote Sensing Ecological Index, Carbon Sink Correction Index" does not seem to be correct. Because the "equivalent factor method" is the method and the last three are indices. And according to your experiment, the quantification and revision of ESV should be done before and after. 4.L13-15:The second sentence of the abstract, "and have ignored the importance of ES spatiotemporal characterization", seems to be disconnected from the first half of the sentence. 5.L158-159:Although you have explained the reason for the revision of the ESV in this sentence, it will still be confusing for the uninitiated reader. Suggest more detail as well as an explanation of the need for the revision. 6.L234-235: On what criteria did the authors base their decision to prioritize the top 10%, 20%, and 30% as identified ecological sources? Literature support or other? 7.L296-321:The results of the spatial transfer of ecological sources are interesting, but a full discussion of this result is lacking in the discussion section of the paper. Above all, it is advised to address the above issues and refine the paper before submitting it. By addressing these points, the manuscript could significantly enhance its contribution to the field of remote sensing and ecological conservation.Author Response
We truly appreciate the reviewer's specific and constructive suggestions on our
manuscript. According to your helpful ideas, we have made considerable corrections and highlighted them in our draft. Please see the attachment.

Reviewer 3 Report
Comments and Suggestions for Authors
Dear Sirs,
as reported in the attached file, I found your paper needing minor improvements, but on the whole interesting for specialist readers, even if its results are not so ground-breaking.
For minor questions I refer you to my notes. I suggest publication after a minor revision. Best regards

In my opinion, the quality of English is on the whole acceptable
Author Response
We truly appreciate the reviewer's specific and constructive suggestions on our
manuscript. According to your helpful ideas, we have made considerable corrections and highlighted them in our draft. Please see the attachment.

Reviewer 4 Report
Comments and Suggestions for Authors
Integrate bibliographic citations in the introduction, there is a lack of similar work done in America or Europe.
In the materials and methods, better specify the types of vegetation and climate of the areas studied.
Improve the quality of the figures, some maps are unclear (perhaps the resolution is too low?).
In the conclusions, explain more about future research developments.
Comments on the Quality of English LanguageA re-reading by a native speaker is recommended
Author Response
Thank you for your kind and constructive comments on our manuscript.We have carefully considered your suggestions and tried our best to improve and made some changes in our manuscript. And the full text has been revised carefully for another time.Please see the attachment.

Reviewer 5 Report
Comments and Suggestions for Authors
The paper presents a methodology approach including landscape aesthetics, carbon sink characteristics, and ecological quality to identify ecological sources from 2000 to 2020, in the Quinghai-Tibetan Plateau, China.
The manuscript is well-written, presented, and documented. I have only a couple of comments.
· A list of acronyms and their explanation would facilitate the reader.
· Applying the proposed methodology led to valuable results for the region. What do you recommend according to your analysis for policymakers and the stakeholders involved in biodiversity conservation at local and global levels?
Author Response
Thank you for your constructive comments and recognition of our manuscript. We
have carefully considered your suggestions and tried our best to improve and make some changes to our manuscript.Please see the attachment.
